# Rethinking Brain-to-Image Reconstruction: What Should We Decode from fMRI Signals?

## Abstract

Recently, notable advancements have been achieved in brain-to-image reconstruction. However, the assumption that the recorded brain activities faithfully mirror the complete high-resolution images conflicts with the workings of human vision and cognitive systems. In this study, we present a novel approach, **fMRI-to-fovea**ted image (**FitFovea**), which redefines the brain-to-image reconstruction process to better align with cognitive science principles. FitFovea comprises three key stages: pseudo-foveated image synthesis, fMRI-to-foveated image **reconstruction** and stimulus image **generation**. In the first stage, FitFovea constructs new {fMRI, pseudo-foveated image} pairs from existing fMRI-image data using saliency prediction and foveated rendering techniques. Next, during the foveated image reconstruction phase, the information captured by human vision is decoded from fMRI signals with maximum accuracy. The final stage, stimulus image generation, is considered not as a strict reconstruction but rather as a postprocessing step. This stage is akin to existing brain-to-image decoding methods, which often emphasize semantic fidelity rather than pixel-level reconstruction. To validate our approach, we introduce the brain score metric to quantify the correlation between images and corresponding brain responses. The superior results validate the rationale behind decoding pseudo-foveated images from fMRI data and demonstrate the feasibility of our newly-devised pipeline based on synthesized pseudo-foveated image training data.

## 1 Introduction

Deciphering the consciousness of the human brain has long been a dream of humanity. Today, propelled by neuroimaging technologies such as functional magnetic resonance imaging (fMRI) and artificial intelligence models like diffusion models (Ho et al., 2020; Rombach et al., 2022), stimulus images of significantly higher quality than ever before have been generated from brain activities (Ozcelik & VanRullen, 2023; Lu et al., 2023; Scotti et al., 2024; Takagi & Nishimoto, 2023). These methods typically adopt the following paradigm: initially, mapping an fMRI signal to the corresponding image feature in the latent space; subsequently, reconstructing the stimulus image based on the obtained image feature utilizing a strong generative model. Implicit in this framework is the foundational assumption that the recorded brain activities faithfully capture the image in its entirety. However, this assumption stands in contrast to prevailing theories regarding the "limited capacity of perceptual experience and cognitive mechanisms" (Cohen et al., 2016; 2012; Luck & Vogel, 2013; Scimeca & Franconeri, 2015; Block, 2011) in cognitive science and neuroanatomy. A prime example can be found in our visual system, where evolution has crafted an elegant balance between maximizing visual perception and minimizing neural resources (Perry & Geisler, 2002). Through the utilization of a foveated retina, a large field of view is encoded at various resolutions, with the central fovea experiencing the highest resolution (as shown in Figure 1). This indicates that the rich visual details in a high-resolution natural scene image can hardly be perfectly encoded in neural signals.

In light of this, a crucial query arises: what should be decoded from fMRI signals in the context of brain-to-image reconstruction? Is the direct decoding of the stimulus image from brain activities the most appropriate choice? Our position leans towards the negative. There is an inherent information gap between fMRI signals and stimulus images (see Appendix A.1 for more theoretical analysis), and striving to link the two could hinder the alignment between these two modalities. To solve this

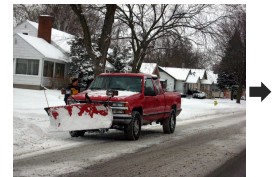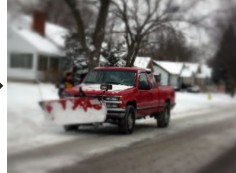

Figure 1: Visual comparison between normal images (left side of the arrows) and pseudo-foveated images (right side of the arrows) with central fixation.

problem, we opt to take a step back and redirect the decoding target from stimulus images to foveated images. Based on this, we present a novel approach, **f**MRI-**t**o-**fovea**ted image (**FitFovea**), which redefines the brain-to-image reconstruction process to better align with cognitive science principles.

FitFovea comprises three key stages: 1) pseudo-foveated image synthesis, 2) fMRI-to-foveated image **reconstruction**, and 3) stimulus image **generation**, as depicted in Figure 2. Addressing the challenge of no training data due to the scarcity of real foveated images, FitFovea proposes the creation of paired {fMRI, pseudo-foveated image} data based on existing {fMRI, image} pairs from the Natural Scene Dataset (NSD) (Allen et al., 2022). To achieve this, saliency prediction models and foveated rendering techniques are employed to generate pseudo-foveated images. Regarding fMRI-to-foveated image reconstruction, the goal is not strictly defined as mere reconstruction. Image representations (essential for verification experiments) or reconstructed pseudo-foveated images are output as needed at this stage. To this end, an autoencoder is utilized to encode pseudo-foveated images into latent representations, enabling the learning of a mapping function to predict pseudo-foveated image representations from fMRI activities. These representations can be further fed into the autoencoder's decoder for pseudo-foveated image reconstruction. While it can be argued that the brain decoding process in this study concludes with the output of the pseudo-foveated image, whether in the form of its embedding or the image itself, the generation of stimulus images is retained as a postprocessing step. This enables the creation of images with consistently high resolution across pixels. Similar to existing brain-to-image decoding methods, achieving pixel-level reconstruction at this stage is often challenging due to incomplete guidance from brain signals. Therefore, this stage emphasizes maintaining semantic fidelity for the generated images. Notably, established brain-to-image reconstruction approaches can be seamlessly integrated with our method at this stage, facilitating the creation of stimulus-related images.

To support our argument, in addition to common metrics for brain-to-image reconstruction, we incorporate the brain score metric introduced by Schrimpf et al. (2018; 2021) to evaluate the correlation between images and brain activities. The superior results of our FitFovea not only demonstrate the rationale for decoding pseudo-foveated images rather than normal images from brain activities but also confirm feasibility of our pipeline built on the constructed pseudo-foveated images.

The main contributions of this work are summarized as follows:

- Rethinking existing brain-to-image reconstruction and developing a novel pipeline Fit-Fovea, a method tailored to decode visual information from brain responses in a manner that aligns more closely with human perceptual and cognitive systems. This approach offers an insightful perspective on the entire process of brain-to-foveated image decoding.

- Introducing a pseudo-foveated image synthesis method that combines algorithms from other fields, i.e. saliency prediction and foveated rendering.

- Exploring individual differences in gaze behavior by synthesizing pesudo-foveated images with fixation points at varying time intervals for each subject, allowing us to identify which time intervals are most strongly related to different subjects' brain activities.

- Adopting a new evaluation metric, brain score, to validate the rationale and feasibility of our approach. Conducting extensive experiments across various backbones, investigating different mapping methods such as ridge regression and MLPs, as well as different generative models, including VAEs and Diffusion models.

## 2 BACKGROUND

**Neural decoding.** Bialek et al. (1989) took an initial step towards recovering the stimulus by decoding the spike train in 1989. This pivotal moment signified a transition for researchers from encoding familiar stimuli to interpreting the neural code. Subsequently, some studies have successfully decoded more intricate stimuli, such as motion direction (Kamitani & Tong, 2005; 2006) and object categories (Haxby et al., 2001; Cox & Savoy, 2003), from recorded functional magnetic resonance imaging (fMRI) signals. More remarkably, Stanley et al. (1999) have attempted to reconstruct movie frames from responses obtained from the lateral geniculate nucleus (LGN). Due to limitations in data and decoding technique, the reconstructed images appear somewhat blurred. Recently, significant progress in cutting-edge artificial intelligence generative models such as variational autoencoders (VAEs) (Van Den Oord et al., 2017; Child, 2020), generative adversarial networks (GANs) (Goodfellow et al., 2014) and diffusion models (Ho et al., 2020; Rombach et al., 2022) has enabled the decoding of stimulus images with unprecedented clarity from brain activities(Lin et al., 2022; Gu et al., 2024; Ozcelik & VanRullen, 2023; Lu et al., 2023; Scotti et al., 2024; Takagi & Nishimoto, 2023; Fang et al., 2024). Beyond visual reconstruction, there is a burgeoning interest in investigating the decoding of linguistic (Makin et al., 2020; Zou et al., 2022; Proix et al., 2022) information from neural signals.

**Foveation and pseudo-foveated image synthesis.** In the human eye, the number of photoreceptors diminishes swiftly from the fovea to the periphery (Curcio et al., 1990). This phenomenon of diminishing photoreceptor density, coupled with an increase in eccentricity, is termed *foveation* (Guenter et al., 2012). Some previous studies explore foveation without the use of eye tracking (Funkhouser & Séquin, 1993; Yee et al., 2001), while others utilize eye tracking hardware (Duchowski, 2002) or foveated displays (Reingold et al., 2003; Duchowski & Çöltekin, 2007). According to the principle of foveation, several studies (Funkhouser & Séquin, 1993; Perry & Geisler, 2002; Viola et al., 2004; Freeman & Simoncelli, 2011; He et al., 2014; Patney et al., 2016; Kaplanyan et al., 2019; Meng et al., 2020; Li et al., 2021; Harrington et al., 2023) delve into pseudo-foveated image/video synthesis based on fixation points within the images or videos. Perry & Geisler (2002) employ an image encoding method using a multi-resolution pyramid, facilitating real-time variable resolution displays. Harrington et al. (2023) utilize the Texture Tilling Model to construct the COCO-Periph dataset, which stands as one of the largest datasets for peripheral vision modeling in deep neural networks. Foveated rendering is the most studied technique among these works, given its vital role in virtual reality. It has the potential to reduce the rendering workload while preserving the user's visual experience. Foveated rendering can be categorized as either fixed (Funkhouser & Séquin, 1993; Viola et al., 2004; Patney et al., 2016) or dynamic(He et al., 2014; Meng et al., 2020; Li et al., 2021), depending on whether the gaze is assumed to be static or dynamic. In this study, we synthesis pseudo-foveated images using fixed foveated rendering. Since this requires input fixation points, we opt to employ current saliency prediction technique, which will be discussed in the following.

**Saliency prediction.** Exploring visual attention is crucial for understanding the human visual system and its application in fields like computer graphics and human-computer interaction (Judd et al., 2009; Chen et al., 2024). One common approach to study human attention is through the utilization of saliency prediction models (Itti et al., 1998; Bruce & Tsotsos, 2005; Harel et al., 2006; Vig et al., 2014; Huang et al., 2015; Bruce et al., 2016; Borji, 2019; Yang et al., 2022; Aydemir et al., 2023), which are designed to detect fixation regions in images or videos. Saliency datasets consists of datasets based on eye tracking data (Judd et al., 2009; Fosco et al., 2020) and those obtained through mouse tracking to simulate eye tracking (Jiang et al., 2015). Early research focused more on static saliency detection, while some recent studies aim to incorporating temporal evolution to generate time-specific saliency (Aydemir et al., 2023). This paper adopts the temporal saliency model (Aydemir et al., 2023), which includes detected saliency in the same duration as the neural decoding dataset utilized in this study, to derive fixation points for pseudo-foveated image synthesis.

## 3 METHOD

FitFovea comprises three primary components: 1) pseudo-foveated image synthesis, 2) fMRI-to-foveated image reconstruction, and 3) stimulus image generation, as depicted in Figure 2. Given the unavailability of real foveated images, we turn to saliency prediction and foveated rendering technologies to simulate foveation and synthesize pseudo-foveated images based on the NSD dataset as

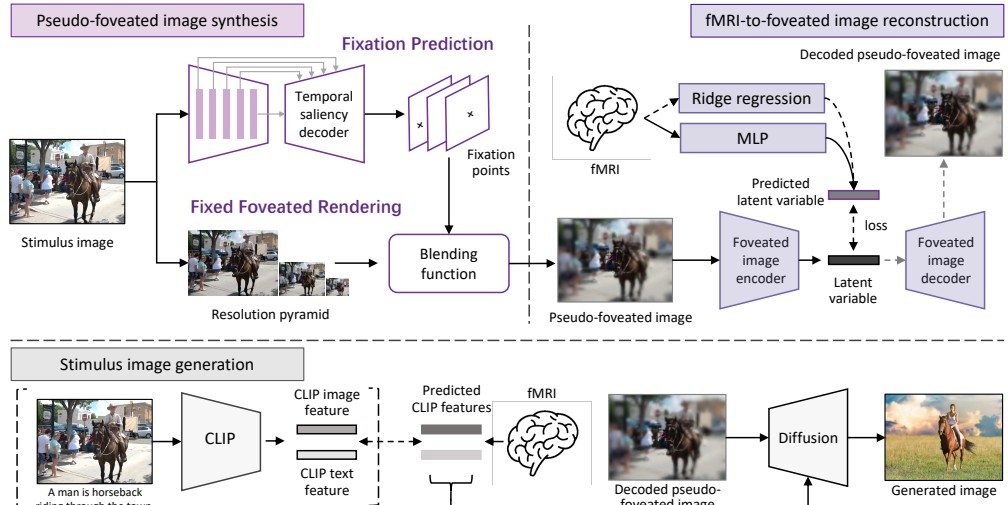

Figure 2: An overview of FitFovea. FitFovea comprises three key stages: 1) pseudo-foveated image synthesis, 2) fMRI-to-foveated image reconstruction, and 3) stimulus image generation. Creation of new {fMRI, pseudo-foveated image} pairs is achieved through saliency prediction and foveated rendering techniques using existing fMRI-image data. An autoencoder is then utilized to encode pseudo-foveated images, and a mapping function is learned to align brain responses with the latent feature space of these images. During the stimulus image generation stage, our model integrates with existing fMRI-to-image reconstruction methods, enabling us to harness the image generative capability of pre-trained diffusion models and facilitate the creation of stimulus-related images.

alternatives. The goal of fMRI-to-foveated image reconstruction is not strictly defined as mere reconstruction. Image representations (essential for verification experiments) or reconstructed pseudo-foveated images are produced as needed at this stage. Finally, the stimulus image generation stage seamlessly integrates our method with existing brain-to-image decoding approaches to create natural scene images. Detailed descriptions of these three parts are provided in the following sections.

## 3.1 PSEUDO-FOVEATED IMAGE SYNTHESIS

### 3.1.1 FIXATION PREDICTION

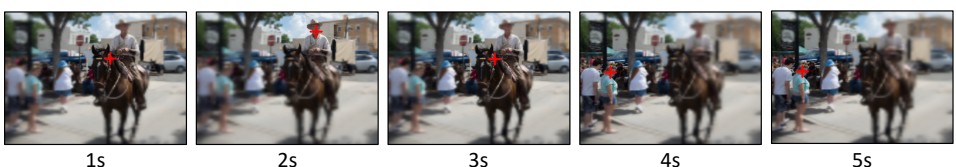

Figure 3: Synthesized pseudo-foveated images based on predicted fixation points (denoted by red crosses) at one-second intervals.

Given that the paired {fMRI, image} data, e.g. NSD (Allen et al., 2022), are collected alongside subjects' image-viewing activities over a defined period, we utilize the temporal saliency prediction model, TempSAL (Aydemir et al., 2023), to produce fixation points. The original TempSAL architecture comprises an image encoder and two saliency decoders: a temporal saliency decoder and a global saliency decoder. With our focus on acquiring fixation points at different time points, the output solely from the temporal saliency decoder suffices. To initiate this process, an image $I$ is input into a pre-trained network, PNASNet-5 (Liu et al., 2018), to extract multi-level features $\mathbf{x}_i$, where $i$ ranges from 1 to 5. Subsequently, the temporal saliency decoder integrates these features through a sequence of four $3 \times 3$ convolutional layers, followed by two additional convolutional layers and a sigmoid function. This finally yields five distinct saliency maps, each corresponding to one-second temporal interval. Within each map, the fixation point is identified as the most salient point, serving as the crucial basis for synthesizing a pseudo-foveated image in subsequent steps. Figure 3 illustrates examples of generated fixation points. In practice, given a 3-second display duration per image in a

scan trial, we retain the first three saliency maps/fixation points representing time intervals of 1s, 2s, and 3s, respectively. One of the three fixation points will be used for synthesizing pseudo-foveated images in the subsequent step.

### 3.1.2 FIXED FOVEATED RENDERING

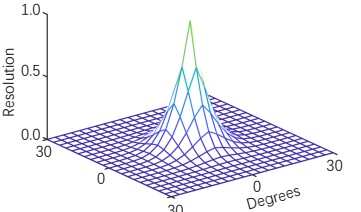 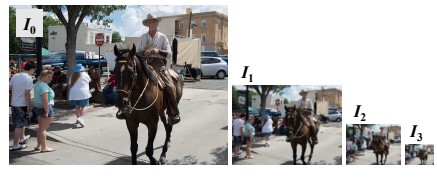

Figure 4: Left: Illustration of a resolution map resembling a normal individual's vision (schematic diagram; see Perry & Geisler (2002) for the resolution map estimated from the "visual fields" using a Goldmann perimeter). Right: First four levels of a multi-resolution pyramid example.

After obtaining the fixation points, we implement the simulation method detailed in Perry & Geisler (2002), known for its efficient processing speed, to transform the original image into a pseudo-foveated form. This method simulates the phenomenon of foveation by gradually reducing resolution in images. It involves generating a series of images with varying resolutions based on the input image and employing a blending function to merge these multi-resolution images into the desired pseudo-foveated image.

Technically, starting with an input image $I$, a multi-resolution pyramid, as depicted in the right portion of Figure 4, is constructed through iterative filtering and down-sampling operations. In Figure 4, the original image $I$ (also $I_0$) represents the initial level. To generate the next level, i.e. image $I_1$, $I_0$ is convolved with a small weighting function, followed by down-sampling of the resulting blurred image in both dimensions. The computation for the remaining levels follows a similar pattern: $I_2$ is derived from $I_1$, $I_3$ from $I_2$, and so forth. Each level within the pyramid corresponds to a specific degree of blur. In our experiments, We employ six pyramid levels, a number commonly deemed sufficient for most applications, as noted in Perry & Geisler (2002). To facilitate the subsequent synthesis stage, the images need to be resized to match the size of the original image via up-sampling and interpolation. These resized images are denoted as $P_i$, with $i = 0, \ldots, 5$.

On the other hand, blending functions are computed based on the predetermined resolution map and the designated fixation point. The original resolution map (an example is depicted in the left portion of Figure 4) is shared among all images. When applied to a specific image, it is adjusted relative to the fixation point to ensure the point aligns with the map's center. Let $R_i$ denotes the fixed spatial resolution corresponding to the $i$-th level of the resolution pyramid, and $B_i(x, y)$ is the blending function for an adjacent pair of $R_i$ and $R_{i-1}$. In Perry & Geisler (2002), Perry and Geisler define a transfer function $f(\cdot)$, from which $B_i(x, y)$ for $R_i < R(x, y) < R_{i-1}$ is derived:

$$B_i(x, y) = \frac{0.5 - f_i(R(x, y))}{f_{i-1}(R(x, y)) - f_i(R(x, y))}, \tag{1}$$

where $R(x, y)$ represents the resolution map function. When $R(x, y) \leq R_i$, $B_i(x, y)$ is set to 0; when $R(x, y) \geq R_{i-1}$, $B_i(x, y)$ is assigned a value of 1. For a six-level pyramid, there are five blending functions, with $B_1$ blending pixels between levels $P_0$ and $P_1$, $B_2$ blending pixels between levels $P_1$ and $P_2$, and so on. The output image $O(x, y)$ is thus defined as:

$$O(x, y) = B_i(x, y)P_i(x, y) + (1 - B_i(x, y))P_{i-1}(x, y). \tag{2}$$

Examples of rendered images corresponding to different fixation points are showcased in Figure 3. The pseudo-foveated images are used to replace the original stimulus images in NSD, forming new {fMRI, pseudo-foveated image} pairs for the subsequent stages.

### 3.2 FMRI-TO-FOVEATED IMAGE RECONSTRUCTION

In this stage, the objective is to decode foveated images from corresponding fMRI activities, either in the form of image embedding or the image itself. The image embedding can be utilized in veri-

fication experiments or for further investigation of normal and foveated images in the future, while the image displays reconstruction outcomes. To this end, autoencoders are employed, which can provide both forms of images through their encoder and decoder architecture. Here we simply select the Stable Diffusion (Rombach et al., 2022) from the diffusion family, and the decoding process is introduced as follows.

During the training phase, the encoder diffusion model begins by taking a pseudo-foveated image as input and produces a latent variable with the dimensions of $4 \times 64 \times 64$, which serves as the pseudo-foveated image embedding. This latent variable acts as the initial point for the decoder diffusion model to reconstruct the input image. Concurrently, a mapping function is trained to convert fMRI data to pseudo-foveated image embeddings. In this study, we explore two mapping approaches: ridge regression and multilayer perceptrons (MLPs), both of which have been demonstrated to be effective in brain decoding (Ozcelik & VanRullen, 2023; Scotti et al., 2024). During inference, no image is provided and only fMRI data is utilized in the decoding process. The fMRI data is input into the mapping function obtained in the training phase to latent variables. These representations can then be fed into the decoder of the diffusion model to reconstruct the original input image.

**Alternative autoencoders.** While Stable Diffusion is employed in this context, various alternatives are available, such as variational autoencoders (VAEs). For more information see Appendix A.2. The results of employing these two different frameworks are detailed in the experimental section.

### 3.3 STIMULUS IMAGE GENERATION

In this study, the brain decoding process to a certain degree wraps up with the earlier pseudo-foveated image reconstruction phase. The subsequent step of stimulus image generation is perceived as post-processing, enabling the creation of images with consistent high resolution among pixels. This output conforms to standard image generation practices and allows for comparisons with existing brain-to-construction approaches. The output of our fMRI-to-foveated image reconstruction stage can be seamlessly integrated with these methods to harness the image generative potential of pre-trained diffusion models and facilitate the creation of stimulus-related images.

In our experiments, we combine the proposed method with two cutting-edge approaches (Ozcelik & VanRullen, 2023; Scotti et al., 2024) due to their good performance in image generation. These methods utilize predicted CLIP features (image and text features) and a middle image containing structure information as inputs to a pretrained diffusion model for image creation. We substitute their middle image input with the output image from our autoencoder's decoder, keeping other settings unchanged. By using the reconstructed pseudo-foveated images or their embeddings, our approach can be intergrated with various existing reconstruction models, circumventing the need to replicate studies in this phase.

## 4 EXPERIMENTS

We use the Natural Scenes Dataset (NSD) (Allen et al., 2022), one of the largest vision-brain datasets, for all experiments. NSD comprises whole-brain 7T fMRI of eight subjects, each exposed to 9000-10000 images from the MSCOCO (Lin et al., 2014) dataset. Each of the eight subjects underwent a unique viewing experience of 9000 images, along with a shared pool of 1000 images that served as the test set. During the fMRI scanning sessions, subjects were presented with images using a design of 4-s trials (3-s ON/1-s OFF) and they needed to judge whether the presented image had been encountered previously. Follow prior studies (Ozcelik & VanRullen, 2023; Takagi & Nishimoto, 2023; Scotti et al., 2024), we conduct experiments on four out of the eight subjects while adhering to the same train/test split. Unless otherwise specified, the results presented are averages across the four subjects. We utilize Stable Diffusion and the very deep VAE (VDVAE) model (Child, 2020) as the autoencoder for our experiments. For more information see Appendix A.2.

### 4.1 RATIONALITY OF DECODING FOVEATED IMAGES FROM FMRI

To validate the rationality of decoding foveated images from fMRI rather than decoding normal images, we adopt the brain score metric introduced by Schrimpf et al. (2018), which evaluates the similarity between an image embedding and the corresponding fMRI scan. To compute the

Table 1: Brain scores for latent variable-to-fMRI prediction. "Normal" denotes normal images, while "Foveated" refers to pseudo-foveated images.

| Method | Stable Diffusion | | VDVAE | | | | | | |
| | MLP | Ridge | layer 1 | layer 2 | layer 3 | layer 4 | layer 5 | avg | all |
|---|---|---|---|---|---|---|---|---|---|
| Normal | .282 | .213 | .162 | .100 | .169 | .125 | .131 | .137 | .201 |
| Foveated | **.297** | **.230** | **.164** | **.108** | **.171** | **.132** | **.145** | **.144** | **.210** |

brain score, an encoder is first utilized to produce an image embedding for an image. This source embedding is then mapped to the target voxels for predicting the brain response $\mathbf{y}_i'$ using ridge regression. Subsequently, the predicted response is compared to the groundtruth response $\mathbf{y}_i$ by calculating the Pearson correlation coefficient $r$:

$$r = \frac{\sum_{i=1}^{n}(\mathbf{y}_i - \bar{\mathbf{y}})(\mathbf{y}_i' - \bar{\mathbf{y}}')}{\sqrt{\sum_{i=1}^{n}(\mathbf{y}_i - \bar{\mathbf{y}})^2}\sqrt{\sum_{i=1}^{n}(\mathbf{y}_i' - \bar{\mathbf{y}}')^2}}. \tag{3}$$

We perform comparative experiments on pseudo-foveated images and normal images to determine which group of images has more brain-like latent variables. The regression coefficient is estimated using the training data and then applied to the test data to compute the brain score. We adopt the encoders described in section 3.2 to encode images (see Appendix A.2 for more details).

Specifically, we compare two encoder models: Stable Diffusion and VDVAE. The results are depicted in Table 1. For Stable Diffusion, we employ two mapping functions: MLPs and ridge regression. We observe relative improvements of 5.3% for MLP and 8.0% for ridge regression in predicting brain responses based on pseudo-foveated images. The MLP mapping method yields higher brain scores. Regarding the latent variables of VDVAE, we report the scores for the first five layers individually and their average. Additionally, we employ the concatenated latent variable from the first 31 layers to predict voxels, following the method in Ozcelik & VanRullen (2023), and present the outcomes in the "all" column. As expected, pseudo-foveated images yield higher scores when utilizing the latent variables either independently or collectively.When used collectively, the brain score rises from 0.201 to 0.210, indicating a relative improvement of 4.5%. These results suggest that pseudo-foveated images exhibit a stronger correlation with fMRI activities compared to normal images, thus validating the rationality of our method.

## 4.2 FIXATION POINT SELECTION FOR PSEUDO-FOVEATED IMAGE SYNTHESIS

Since we lack real foveated image data, it is crucial to identify the pseudo-foveated image (synthesized with fixation points at varying time intervals) that closely resembles a real one. Given the variability in gaze behavior among individuals, we conduct experiments for each subject. The evaluation metric aligns with that used in Section 4.1. We present the results for subject 01 in Figure 5, employing VDVAE as the image encoder and mapping latent variables to fMRI using ridge regression. Within each bar chart set, progression from

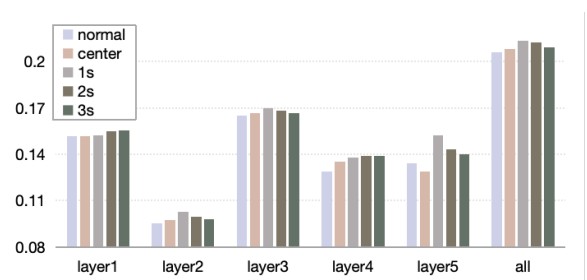

Figure 5: Comparison of brain scores for normal images, pseudo-foveated images synthesized with central fixation (center), and with fixation points predicted at different time intervals (1s, 2s, 3s).

left to right represents the normal image, the pseudo-foveated image synthesized based on central fixation, and those with predicted fixation points at 1, 2, and 3 seconds. The results of the "all" group are given the most weight when selecting the fixation point for a subject. The results in Figure 5 indicates that the pseudo-foveated image at the first second exhibits the highest correlation with the corresponding brain response. Furthermore, experimenting with our synthesized image, regardless of using fixation points from 1, 2, or 3 second, outperforms the normal image or using central fixation. For results pertaining to other subjects, please refer to Appendix A.3. The results of

the remaining three subjects all highlight the 2-second images. When aggregating results across all four subjects, it appears that during a 3-second image display, content attracting early to mid-level attention is more likely to be reflected in fMRI scans.

## 4.3 FMRI-TO-FOVEATED IMAGE RECONSTRUCTION

Table 2: Pearson correlation coefficient results for fMRI-to-latent variables prediction. "Normal" denotes normal images, while "Foveated" refers to pseudo-foveated images.

| Method | Stable Diffusion | | VDVAE | | | | | |
|---|---|---|---|---|---|---|---|---|
| | MLP | Ridge | layer 1 | layer 2 | layer 3 | layer 4 | layer 5 | avg |
| Normal | .370 | .335 | .595 | .433 | .372 | .433 | .237 | .414 |
| Foveated | **.542** | **.464** | **.797** | **.704** | **.379** | **.461** | **.439** | **.556** |

To further assess the quality of fMRI decoding, we calculate the Pearson correlation coefficient between the predicted latent variables from fMRI and the target latent variables generated by feeding the corresponding image into an image encoder. This process can be seen as a reverse operation compared to brain score analysis. Again, the encoders utilized are Stable Diffusion and VDVAE, with an exploration of voxel mapping to either the embedding space of normal or pseudo-foveated images. Detailed results are presented in Table 2. When mapping fMRI data to the latent space of Stable Diffusion using MLP, the result improves from 0.37 for normal images to 0.542 for pseudo-foveated images, reflecting a 46.5% relative increase. Utilizing ridge regression yields a relative improvement of 38.5%. A similar pattern emerges with VDVAE. For layers 1 and 2 of VDVAE, a significant performance boost is observed when transitioning to predicting pseudo-foveated images, likely due to the lower dimensionality of the latent variables than deeper layers. The average correlation across layers 1-5 is 0.414 for normal images and 0.556 for pseudo-foveated images, showcasing a 34.3% relative improvement. This time, the prediction of the concatenated latent variable for VDVAE is omitted due to the substantial challenge posed by its high dimensionality of 91168. The increased correlation not only indicates enhanced prediction accuracy for the embeddings of specific image type but also suggests the predictability of this image type. These findings reinforce the rationale for decoding foveated images from fMRI rather than decoding normal images.

For evaluation of reconstructed pseudo-foveated images, we adopt the evaluation metrics outlined in Scotti et al. (2024); Ozcelik & VanRullen (2023). This involves assessing the methods using low-level metrics such as pixel-wise correlation (PixCorr), and the structural similarity index metric (SSIM), as well as high-level metrics like the average correlation distances of EfficientNet-B1 (Eff) and SwAV-ResNet50 (SwAV). Furthermore, two-way identification based on the output embeddings of AlexNet (Krizhevsky et al., 2012) (the second layer and the fifth layer), Inception V3 (Szegedy et al., 2016) (last pooling layer), and CLIP (Radford et al., 2021) (final layer of ViT-L/14) is also performed. Table 3 and Table 4 provide a detailed examination of the performance of output images generated by the Stable Diffusion decoder. In Table 3, the evaluation is based on calculating the metrics between the generated images and synthesized pseudo-foveated images. Superior perfor-

Table 3: Comparison of normal (-Normal) and pseudo-foveated (-Foveated) image reconstruction in the **fMRI-to-foveated image reconstruction** stage. The results are evaluated by computing the metric between the generated images and **pseudo-foveated** images. (S1 denotes subject 01; if marked as "Ridge", ridge regression is emplyed, otherwise, MLPs are used; see Appendix A.4 for detailed individual subject results.)

| Method | Low-Level | | | | High-Level | | | |
|---|---|---|---|---|---|---|---|---|
| | PixCorr ↑ | SSIM ↑ | Alex(2) ↑ | Alex(5) ↑ | Incep ↑ | CLIP ↑ | Eff ↓ | SwAV ↓ |
| SD-Normal-S1(Ridge) | .404 | .546 | 84.4% | 74.9% | 56.3% | 52.7% | .935 | .570 |
| SD-Foveated-S1(Ridge) | **.407** | **.568** | **84.7%** | **77.9%** | **57.4%** | **52.8%** | **.906** | **.543** |
| SD-Normal-S1 | .471 | .629 | 82.4% | 76.9% | 58.2% | 55.1% | .890 | .557 |
| SD-Foveated-S1 | **.482** | **.636** | **90.3%** | **90.4%** | **66.4%** | **63.8%** | **.874** | **.524** |
| SD-Normal | .401 | .621 | 82.4% | 81.9% | 63.7% | 59.5% | .881 | .525 |
| SD-Foveated | **.405** | **.628** | **85.1%** | **85.8%** | **65.6%** | **62.1%** | **.873** | **.524** |

Table 4: Comparison of normal (-Normal) and pseudo-foveated (-Foveated) image reconstruction in the **fMRI-to-foveated image reconstruction** stage. The results are evaluated by calculating the metrics between the generated images and **groundtruth stimulus** images. (See Appendix A.4 for detailed individual subject results.)

| Method | Low-Level | | | | High-Level | | | |
|---|---|---|---|---|---|---|---|---|
| | PixCorr ↑ | SSIM ↑ | Alex(2) ↑ | Alex(5) ↑ | Incep ↑ | CLIP ↑ | Eff ↓ | SwAV ↓ |
| SD-Normal-S1(Ridge) | .385 | .432 | 83.7% | 73.7% | 55.3% | 52.8% | .995 | **.662** |
| SD-Foveated-S1(Ridge) | **.386** | **.438** | **84.0%** | **74.5%** | **55.8%** | **53.0%** | **.992** | .664 |
| SD-Normal-S1 | .456 | **.493** | 87.1% | 84.1% | 61.6% | 62.4% | .992 | .638 |
| SD-Foveated-S1 | **.464** | .489 | **89.1%** | **89.5%** | **65.9%** | **66.3%** | **.975** | **.621** |
| SD-Normal | .360 | **.479** | 78.1% | 74.8% | 58.7% | 59.2% | 1.00 | .663 |
| SD-Foveated | **.391** | .478 | **83.4%** | **83.4%** | **63.5%** | **63.7%** | **.984** | **.623** |

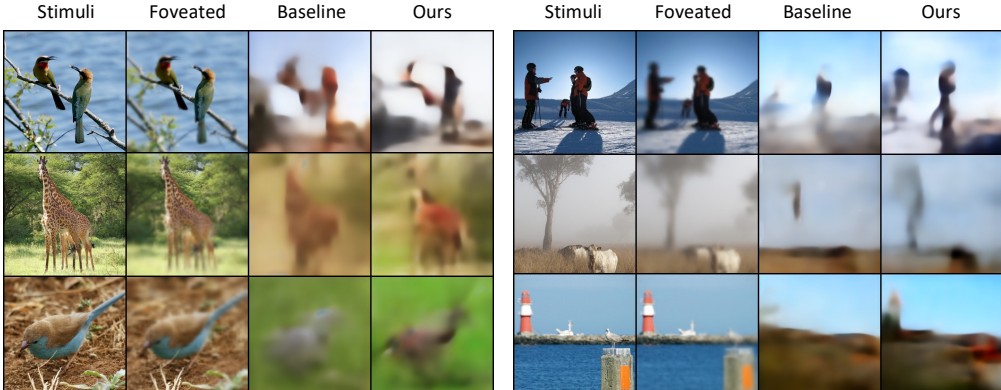

Figure 6: Exemplary reconstructed pseudo-foveated images (Ours) and normal images (Baseline) from the fMRI data of subject 01. The first two columns showcase stimulus images (Stimuli) and pseudo-foveated images (Foveated) synthesized in Section 3.1.

mance is exhibited when decoding pseudo-foveated images compared to normal images across all low-level and high-level metrics. Table 4 focuses on computing the metrics between the generated images and groundtruth stimulus images, showing superior performance in most metrics. Example reconstructed pseudo-foveated images can be found in Figure 6, where "Baseline" refers to normal image reconstruction and "Ours" denotes pseudo-foveated image reconstruction. The first two columns showcase stimulus images (Stimuli) and pseudo-foveated images (Foveated). In comparison to the baseline outcomes, key objects in the reconstructed pseudo-foveated images are clearer, with more accurate shapes, positions and details.

## 4.4 RESULTS OF STIMULUS IMAGE GENERATION

Based on BrainDiffuser (Ozcelik & VanRullen, 2023) and MindEye (Scotti et al., 2024), we can finally complete the image generation process. In Figure 7, exemplary images created by Brain-Diffuser, MindEye, and the combination of our FitFovea with MindEye (Ours) using fMRI signals of subject 01 are showcased. By incorporating our generated pseudo-foveated images, the stimulus images produced by our method exhibit enhanced structural and positional information. Additional images created from fMRI signals of all subjects see Appendix Figure 8 and 9. To quantitatively contrast with other methods, we present the results of comparing the generated images with the groundtruth stimulus images in Table 5. It is exciting to discover that, when combined with our Fit-Fovea, BrainDiffuser and MindEye both demonstrate improved performance overall. This could be attributed to the more precise visual information captured by FitFovea being successfully conveyed in the generated results. Furthermore, our methodology demonstrates significant enhancements in low-level metrics such as PixCorr and SSIM. This suggests that FitFovea excels more in pixel-level reconstruction, which is paramount in brain-to-image reconstruction.

Figure 7: Exemplary images produced by BrainDiffuser (Ozcelik & VanRullen, 2023) , MindEye (Scotti et al., 2024), and MindEye+FitFovea (Ours) using fMRI signals of subject 01. The first two columns showcase stimulus images (Stimuli) and pseudo-foveated images (Foveated).

Table 5: Comparison of the performance in **stimulus image generation** using FitFovea combined with two brain-to-image reconstruction methods against other models. The results are based on computing the metrics between the generated images and **groundtruth stimulus** images.

| Method | Low-Level | | | | High-Level | | | |
|---|---|---|---|---|---|---|---|---|
| | PixCorr ↑ | SSIM ↑ | Alex(2) ↑ | Alex(5) ↑ | Incep ↑ | CLIP ↑ | Eff ↓ | SwAV ↓ |
| MindReader (Lin et al., 2022) | – | – | – | – | 78.2% | – | – | – |
| LDM (Takagi & Nishimoto, 2023) | – | – | 83.0% | 83.0% | 76.0% | 77.0% | – | – |
| Cortex2Image (Gu et al., 2023) | .150 | .325 | – | – | – | – | .862 | .465 |
| BrainDiffuser (Ozcelik & VanRullen, 2023) | .254 | .356 | **94.2**% | 96.2% | 87.2% | 91.5% | .775 | **.423** |
| BrainDiffuser+ours | **.264** | **.360** | **94.2**% | **96.6**% | **89.7**% | **91.8**% | **.750** | .429 |
| MindEye (Scotti et al., 2024) | .309 | .323 | 94.7% | **97.8**% | 93.8% | 94.1% | .645 | **.367** |
| MindEye+ours | **.327** | **.342** | **95.1**% | 97.5% | **94.3**% | **94.4**% | **.642** | .372 |

## 5 CONCLUSION

In this study, we have redesigned the brain-to-image reconstruction process by drawing on insights from human cognitive science and neuroanatomy, leading to the development of a novel reconstruction pipeline called FitFovea. FitFovea effectively addresses the issue of lack of training data and achieves fMRI-to-foveated image reconstruction based on synthesized pseudo-foveated images. The experimental findings validate the rationale for decoding pseudo-foveated images from brain activities and proves the viability of our decoding approach. We aspire that our reconsideration of brain-to-image reconstruction and the introduction of FitFovea will stimulate further exploration in this field, encouraging the development of decoding and reconstruction networks that better emulate human brain processing, and further propelling the integration and progression of cognitive science and artificial intelligence.

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

# A APPENDIX

## A.1 MORE THEORETICAL ANALYSIS

The retina is the initial site for capturing visual information, converting light into neural signals transmitted to the brain (Guilherme & Leon, 1999). According to information theory (Shannon, 1948; Shannon & Weaver, 1963), specifically the data-processing inequality (Shannon & Weaver, 1963), information sent through noisy communication channels experiences inevitable loss that cannot be recaptured through further processing. This principle suggests that the visual information processed by the visual systems is constrained by what the retina initially captures. Besides, the anatomical structure of the retina (Curcio & Allen, 1990; Curcio et al., 1990) determines inherent differences between the fovea and peripheral regions, with resolution peaking at the fovea and declines toward the periphery (Curcio et al., 1990). Statistical analyses also support this observation (Cohen et al., 2016; Freeman & Simoncelli, 2011). The blur and distortion in the periphery of foveated images (Pointer & Hess, 1989; Stewart et al., 2020) signify a noteworthy reduction in information compared to the original stimuli. Given that the maximum visual information conveyed by brain acitivity cannot exceed that captured by the retina, we surmise that foveated images correlate more strongly with brain activity than the original stimuli.

## A.2 MORE DATA INFORMATION AND EXPERIMENTAL DETAILS

Following previous studies (Ozcelik & VanRullen, 2023; Takagi & Nishimoto, 2023; Scotti et al., 2024), we conduct experiments involving four subjects who completed all imaging sessions—subjects 01, 02, 05, and 07. Each subject was exposed to different training images, while the test set remained consistent. Each subject underwent three scan sessions for each image. Our experiments involve averaging brain activities for the test set images following Takagi & Nishimoto (2023); Scotti et al. (2024), while data from all scan sessions for training images are separately used. For the fMRI-to-foveated image reconstruction stage, models are trained using a single A100. During the stimulus image generation stage, we directly use the model weights provided by Ozcelik & VanRullen (2023); Scotti et al. (2024).

When using VDVAE as the alternative autoencoder in Section 3.2, during the training phase, a pseudo-foveated image is first fed into the 75-layer encoder to generate a group of latent variables from bottom to up. Due to the high dimensionality of the latent variables from all layers, we only employ ridge regression as the mapping approach. We take two mapping strategies. Firstly, the fMRI data is individually mapped to the latent variable of each layer, achieved by training a distinct ridge regression model for each mapping. Secondly, the fMRI data is mapped to the concatenation of latent variables of 31 layers, this forms a variable of 91168 dimension and only a single ridge regression model is required in this case. During inference, no image is provided and only fMRI data is utilized in the decoding process. The fMRI data is input into the ridge regression models obtained in the training phase to produce a series of latent variables or a variable for 31 layers together. These representations can then be fed into the decoder of VDVAE to reconstruct the original input image. When using Stable Diffusion, we experiment with both ridge regression and MLPs as mapping functions. In the case of MLPs, the fMRI voxels are initially processed by an MLP, producing an output with dimensions of $64 \times 16 \times 16$, which is then upsampled to align with the dimension of the latent variable. The MLP is trained using the mean squared error (MSE) loss between the predicted and target latent variables.

For the experiments conducted in Section 4.1, we utilize the pre-trained VDVAE as detailed in Child (2020), without any further fine-tuning. Thus the same VDVAE model is utilized for encoding both normal images and pseudo-foveated images. When employing Stable Diffusion for image encoding, we finetune the pre-trained model with normal images before extracting embeddings for normal images. Additionally, for pseudo-foveated images, we finetune it with pseudo-foveated images before extracting the corresponding embeddings.

## A.3 SUBJECT-SPECIFIC FIXATION POINT SELECTION

Here, we present the brain scores for latent variable-to-fMRI prediction and the Pearson correlation coefficient results for fMRI-to-latent variable prediction in Table 6, individually for each subject.

Table 6: Comparison of brain scores for normal images, pseudo-foveated images synthesized with central fixation (center), and with fixation points predicted at different time intervals (1s, 2s, 3s), reported for each subject. Pearson correlation coefficient results for fMRI-to-latent variables prediction are also presented in the right portion.

| Subject | Image | Latent to fMRI | | | | | | fMRI to latent | | | | |
|---|---|---|---|---|---|---|---|---|---|---|---|---|
| | | layer 1 | layer 2 | layer 3 | layer 4 | layer 5 | all | layer 1 | layer 2 | layer 3 | layer 4 | layer 5 |
| Subj01 | normal | .152 | .095 | .165 | .129 | .134 | .206 | .613 | .453 | .402 | .499 | .269 |
| | center | .152 | .098 | .168 | .135 | .129 | .208 | .783 | .702 | .405 | .526 | .462 |
| | 1s | .152 | **.103** | **.170** | .138 | **.152** | **.214** | .798 | .712 | .406 | .525 | **.469** |
| | 2s | .155 | .099 | .168 | **.139** | .143 | .213 | .808 | .719 | **.408** | **.528** | .457 |
| | 3s | **.156** | .098 | .167 | **.139** | .140 | .209 | **.811** | **.720** | **.408** | .523 | .449 |
| Subj02 | normal | .161 | .099 | .175 | .125 | .130 | .210 | .595 | .430 | .376 | .445 | .247 |
| | center | .163 | .108 | .177 | .132 | .131 | .209 | .772 | .691 | .375 | .473 | **.458** |
| | 1s | .160 | **.111** | .179 | .132 | **.155** | .216 | .789 | .702 | .380 | .473 | .454 |
| | 2s | .164 | .109 | .177 | .134 | .143 | **.221** | .800 | .707 | .381 | **.474** | .441 |
| | 3s | **.166** | .108 | **.180** | **.135** | .140 | .220 | **.804** | **.710** | **.384** | .467 | .440 |
| Subj05 | normal | .199 | .124 | .201 | .141 | .153 | .233 | .603 | .456 | .364 | .392 | .213 |
| | center | .199 | **.134** | **.202** | .147 | .150 | .231 | .776 | .696 | .366 | .420 | **.438** |
| | 1s | .197 | .131 | **.202** | .147 | **.182** | .239 | .793 | .702 | **.372** | **.422** | .435 |
| | 2s | .201 | **.134** | **.202** | .149 | .168 | **.244** | .802 | .711 | **.372** | .421 | .432 |
| | 3s | **.202** | .132 | .201 | **.151** | .162 | .241 | **.808** | **.714** | .370 | **.422** | .421 |
| Subj07 | normal | .136 | .080 | .135 | .104 | .106 | .156 | .569 | .396 | .348 | .395 | .220 |
| | center | .135 | .084 | .138 | .106 | .105 | .153 | .758 | .668 | .355 | .424 | **.434** |
| | 1s | .135 | **.088** | .140 | .105 | **.125** | .158 | .777 | .682 | **.360** | **.426** | **.434** |
| | 2s | **.137** | .085 | .138 | .108 | .118 | **.161** | .788 | .686 | .357 | .425 | .422 |
| | 3s | .136 | .085 | **.141** | **.109** | .115 | .156 | **.792** | **.692** | **.360** | .423 | .413 |

The assessment involves comparing the outcomes based on normal images, pseudo-foveated images synthesized with central fixation and fixation points predicted at different time intervals.

## A.4 SUBJECT-SPECIFIC IMAGE RECONSTRUCTION RESULTS

Table 7: Quantitative results of pseudo-foveated image reconstruction for individual subjects in the **fMRI-to-foveated image reconstruction** stage. Evaluation is based on computing the metrics between the generated images and **pseudo-foveated** images. Aggregated average scores across subjects are shown in Table 3.

| Subject | Low-Level | | | | High-Level | | | |
|---|---|---|---|---|---|---|---|---|
| | PixCorr ↑ | SSIM ↑ | Alex(2) ↑ | Alex(5) ↑ | Incep ↑ | CLIP ↑ | Eff ↓ | SwAV ↓ |
| Subj01 | .482 | .636 | 90.3% | 90.4% | 66.4% | 63.8% | .874 | .524 |
| Subj02 | .421 | .633 | 88.1% | 88.8% | 67.1% | 63.1% | .868 | .521 |
| Subj05 | .368 | .621 | 81.9% | 83.4% | 65.2% | 61.5% | .872 | .528 |
| Subj07 | .350 | .621 | 79.9% | 80.4% | 63.7% | 59.9% | .877 | .525 |

Table 8: Quantitative results of pseudo-foveated image reconstruction for individual subjects in the **fMRI-to-foveated image reconstruction** stage. Evaluation is based on computing the metrics between the generated images and **groundtruth stimulus** images. Aggregated average scores across subjects are shown in Table 4.

| Subject | Low-Level | | | | High-Level | | | |
|---|---|---|---|---|---|---|---|---|
| | PixCorr ↑ | SSIM ↑ | Alex(2) ↑ | Alex(5) ↑ | Incep ↑ | CLIP ↑ | Eff ↓ | SwAV ↓ |
| Subj01 | .464 | .489 | 89.1% | 89.5% | 65.9% | 66.3% | .975 | .621 |
| Subj02 | .406 | .480 | 86.3% | 86.2% | 64.1% | 64.5% | .981 | .619 |
| Subj05 | .355 | .471 | 80.3% | 79.8% | 62.8% | 62.9% | .987 | .627 |
| Subj07 | .339 | .470 | 77.8% | 77.9% | 61.1% | 61.2% | .991 | .626 |

Here we showcase the results of pseudo-foveated image reconstruction for individual subjects in Table 7 and 8.

## A.5    MORE RECONSTRUCTIONS AND GENERATIONS

Additional images created from fMRI signals of all subjects during the stimulus image generation phase can be found in Figure 8 and Figure 9.

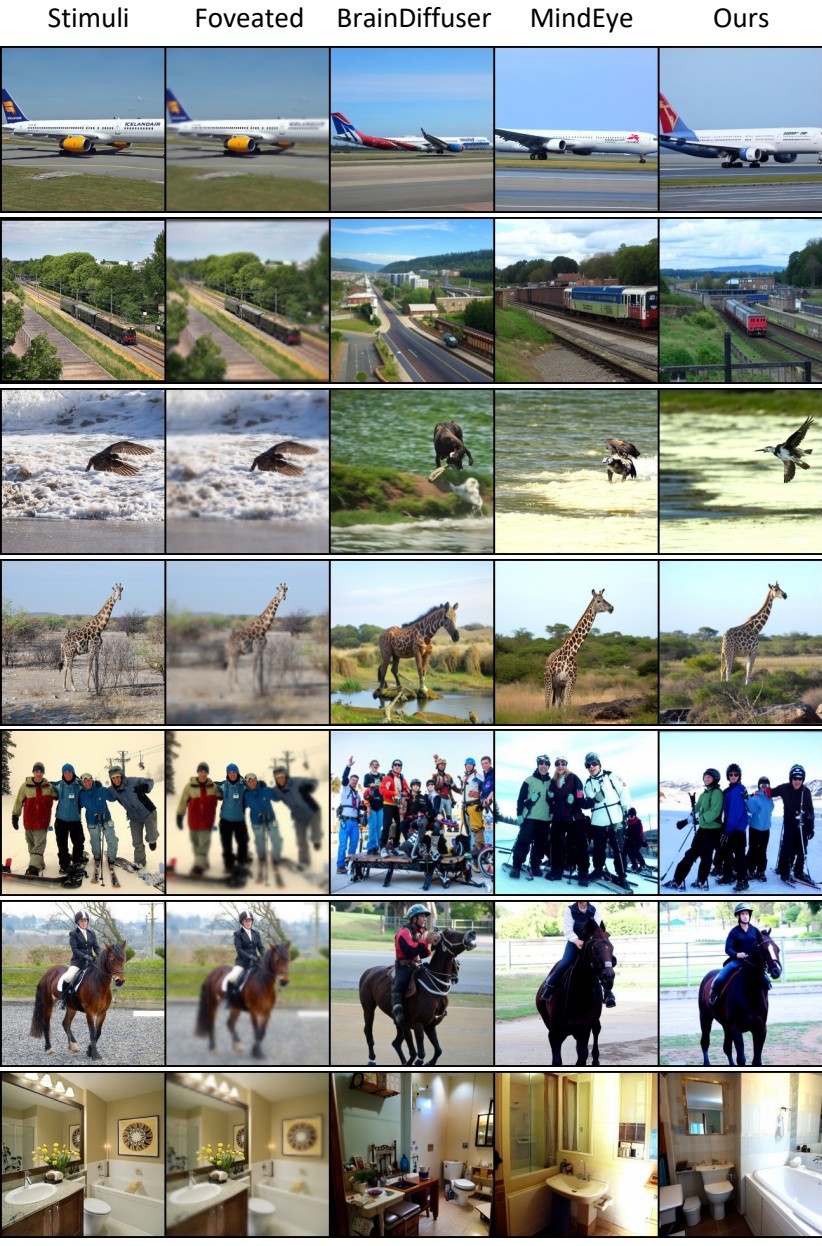

Figure 8: Additional exemplary images produced by BrainDiffuser (Ozcelik & VanRullen, 2023) , MindEye (Scotti et al., 2024), and MindEye+FitFovea (Ours) using fMRI signals of subject 01. The first two columns showcase stimulus images (Stimuli) and pseudo-foveated images (Foveated).

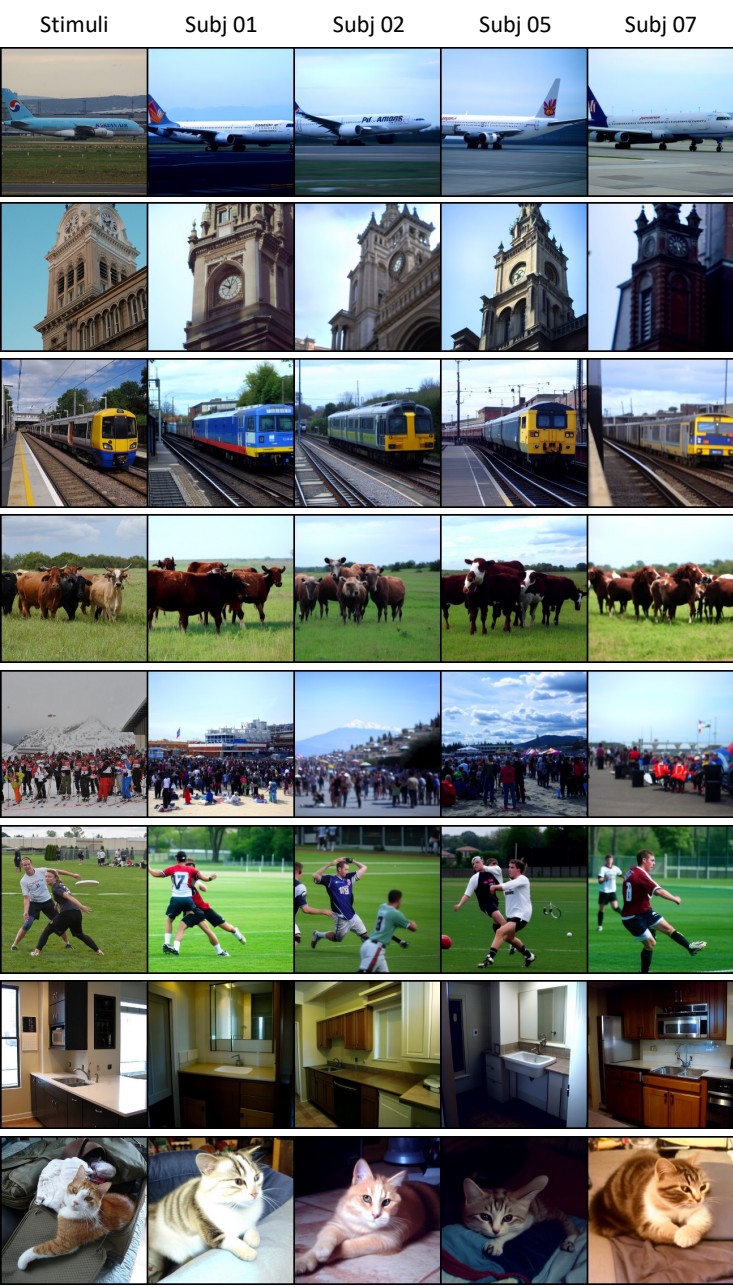

Figure 9: Additional exemplary images produced by our method using the fMRI signal of four subjects. The first column showcases the stimulus images (Stimuli).

