# OpenReview forum: "Rethinking Brain-to-Image Reconstruction: What Should We Decode from fMRI Signals?"
_ICLR.cc/2025/Conference — ICLR 2025 Conference Withdrawn Submission_

### Official Review · Reviewer_Wtv1 · 2024-10-25

**Soundness:** 2
**Presentation:** 3
**Contribution:** 1
**Rating:** 3
**Confidence:** 4

**Summary:**

This paper addresses the challenge of image reconstruction from fMRI recordings. Motivated by the observation that the fMRI signal is likely to provide an incomplete account of the image presented to the subjects, the authors propose a two-step reconstruction approach, where the first is a lower bar reconstruction problem aiming to reconstruct a "pseudo-foveated" version of the ground truth image, and the second is an enhancement thereof based on a strong natural image prior and decoded semantic information guidance. The proposed method is demonstrated on the Natural Scene Dataset, arguing in favor of a pseudo-foveated reconstruction decoupled approach, and shown to confer moderate gains when integrated into previous methods.

**Strengths:**

The authors' exploration of the relationship between fMRI recordings and their potential to approximate ground truth images, suggests the concept of a reconstruction *empirical ceiling*, which I believe is a novel and compelling direction.
This approach questions the prevailing assumption that reconstructions can indefinitely approximate the original stimulus, where ambiguities are in practice often resolved using strong natural image priors.
By addressing the inherent information boundary between fMRI-driven reconstruction and post-processing, this work begins to probe how one might bias the reconstruction process to prioritize fidelity to the fMRI cues.
I appreciate the authors’ initiative in tackling this under-explored dimension, which could contribute valuable insights into the limitations and potential of fMRI-guided image reconstruction.

However, while the concept of an "inherent information gap" between fMRI signals and stimulus images is intriguing, the authors' choice of a pseudo-foveated image as the optimal decoupling point in the decoding process could benefit from further scientific grounding (see Weaknesses). The selection of a foveated image as the intermediary may not fully address the nuanced dimensions of this information gap.

**Weaknesses:**

Given the substantial question of information limits in fMRI recordings, there are several critical components to consider: 1) Since subjects are center-fixated, the early stages of visual processing likely represent a pseudo-foveated version of the image—an incomplete view of the full stimulus; 2) The BOLD signal is an indirect measure of neural activity, contributing to further information loss; 3) The spatial resolution of fMRI (~1,000 neurons per voxel at 7T) imposes additional constraints on the reconstruction ceiling.

Ideally, each of these limitations would be independently assessed, with primary attention to factors like signal granularity and voxel resolution (2-3), which likely dominate information loss. Instead, the authors focus on center-fixation, the factor that may be least influential here, without robustly measuring this impact given current recording limitations. This raises concerns about whether the presented findings are genuinely due to the “limited capacity of perceptual experience and cognitive mechanism” mentioned as a key premise.

A second major issue is the choice of saliency-based foveation. Since NSD subjects were reportedly center-fixating, any resulting neural representation should align with a **center-foveated** version of the stimulus. The use of artificial predictors to estimate general saliency—creating varying foveated versions—seems scientifically unsupported unless subjects were viewing freely. This is a fundamental issue, as it challenges the entire premise of the foveated reconstruction approach in this context. Whether or not this potentially confers some gain to reconstruction quality by a mysteriously emerging optimization/regularization tweak, is another topic, which I address below.

Additional Points for Improvement:

- The paper places excessive emphasis on generating foveated images, with minimal grounding in the literature on this approach’s scientific merit for fMRI reconstruction.
- The review of prior work is sparse, missing both past and recent studies while presenting weakly related works in detail.
- Evaluations are based solely on a single dataset, limiting the generalizability of findings.
- [67-86] Unclear.
- [86-91] BrainScore, typically used for model evaluation, seems unrelated to image reconstruction—its relevance here is unclear.
- [94-97] “Rethinking” without elaboration lacks contribution.
- [101-103] The reference to gaze behavior and time intervals appears abruptly, lacking context.
- [105-108] Unclear use of BrainScore and for a single fMRI dataset, neither fully justified and not comprehensive.
- The brain-score metric, although marked as the fourth contribution, is disproportionately highlighted across figures and tables.
- Table 5, intended as a comparison with prior works, shows only marginal support for the proposed approach, weakening the overall case.

**Questions:**

1) In the NSD dataset, subjects are center-fixating. Given this, if the hypothesis is that the resulting neural representation reflects a limited foveated view of the image, should this not be a **center-foveated** version exclusively?

2) Could the proposed foveation algorithm be compared with a log-polar transformation? This transformation would naturally align with early visual cortex representation, which tends to dominate fMRI reconstructions.

3) The organization of the comparison results is challenging to follow, particularly with varying references to ground truth, foveated versions of ground truth, variants of the authors' approach, and other methods. It would be beneficial to clarify the main findings, highlighting the added value of the approach, especially when contrasting the best final result with those of previous leading approaches.

4) The main results, particularly in Table 3, lack uncertainty estimates, making it difficult to evaluate the significance of the findings. Can uncertainty estimates be added for clarity?

5) In Figure 6, the proposed approach appears to offer visually discernible improvement, yet: A) The example appears cherry-picked; could more examples be added to provide a representative view? B) The "Baseline" seems to represent a variant of the authors' method ("normal") rather than a direct comparison to prior work. Could the baseline instead show results from previous works to allow a clearer benchmark?

6) Figure 7, as a main figure of merit, does not seem to show substantial improvement and may appear cherry-picked. Would the authors consider moving it earlier in the paper and adding more examples to provide a fuller view of performance?

---

### Official Review · Reviewer_3nQF · 2024-10-31

**Soundness:** 3
**Presentation:** 3
**Contribution:** 2
**Rating:** 3
**Confidence:** 4

**Summary:**

The authors propose an extension to brain-to-image reconstruction pipelines by leveraging the evolutionary insight that human vision has developed with a foveated structure. They generate pseudo-foveated images and conduct a series of well-crafted experiments demonstrating that these images exhibit higher similarity to corresponding brain activity compared to non-foveated images. By integrating this approach as an enhancement to the low-level components of existing pipelines, they achieve improved results across most metrics.

**Strengths:**

The approach presented is intriguing. The theoretical analysis of the inherent information gap between fMRI signals and stimulus images is insightful, though it could benefit from further depth. The method for generating pseudo-foveated images is promising and could show significant value if validated with real data. The experiment justifying the selection of the fixation point timing is well-designed and reinforces the authors' argument. The paper is well-structured, with a logical flow and a good selection of schematics and graphs. Overall, it offers a straightforward addition to the low-level pipeline of existing architectures that demonstrates a marginal performance improvement.

**Weaknesses:**

1) The motivation for using TempSAL to generate fixation points is unclear, especially considering that eye-tracking information is already available in the NSD dataset. If one of the goals of this approach is for temporal saliency prediction to serve as a substitute for eye-tracking data, it would be valuable to include an analysis comparing TempSAL’s predictions with the actual gaze positions of subjects. This comparison could help clarify the accuracy and relevance of using TempSAL in place of direct eye-tracking data.

2) The authors' focus on a single-subject pipeline would benefit from a comparison with state-of-the-art methods. Notably, MindEye2 (Scotti et al., 2024 Table 6) has demonstrated superior performance over MindEye1 in single-subject scenarios. A comparative analysis with MindEye2 is essential to validate the proposed method's effectiveness. Additionally, exploring the application of this approach within multi-subject pipelines could provide valuable insights into its broader applicability.

3) The brain has evolved to operate in a ‘foveated’ manner, and the authors have effectively demonstrated a higher similarity between pseudo-foveated image embeddings and fMRI signals. However, since we cannot be certain that the generated fixation point for a given image matches the subject's actual focus during the scanning session, how can we be sure that the increased correlation is not simply due to the inherently higher SNR of foveated image embeddings compared to full-image embeddings? Concretely, foveated images naturally contain fewer peripheral details, which reduces noise in the embeddings. This leads us to expect that the model would produce cleaner, less noisy embeddings, as it does not need to encode as much extraneous peripheral information. As a result, the model’s latent representations are more concentrated on relevant features, which could increase the overall signal-to-noise ratio in the embeddings.

4) Line 101, 421: typo

**Questions:**

Addressing the issues raised in the weaknesses section would significantly enhance the paper. By tackling these points, the authors can strengthen the clarity and robustness of their approach, providing a more comprehensive validation of their findings and addressing any potential limitations. This would make the contribution more convincing and impactful.

---

### Official Review · Reviewer_M6Ap · 2024-11-04

**Soundness:** 2
**Presentation:** 3
**Contribution:** 2
**Rating:** 5
**Confidence:** 3

**Summary:**

This paper proposes a novel brain-to-image reconstruction named FitFovea. Given the information gap between fMRI signals and stimulus images, FitFovea redefines the brain-to-image reconstruction process and combines saliency prediction and foveated rendering to decode visual information from brain responses in a way that more closely resembles human perceptual and cognitive systems. The authors validate the method on the NSD dataset and obtain superior results.

**Strengths:**

1. Proposed a novel brain-to-image reconstruction method by considering the conflicts with the workings of human vision and cognitive systems.
2. FitFovea considers the unavailability of real foveated images and aims to decode foveated images from corresponding fMRI activities, and integrates the aforementioned method with existing brain-to-image decoding approaches to create natural scene images.
3. The authors validate the proposed algorithm on the NSD dataset and obtain superior results.

**Weaknesses:**

1. Why choose ridge regression and MLP to encode fMRI data instead of CNN? Convolution of images is a state-of-the-art method.
2. What is the low level in Table 3 and Table 4?
3. What is the meaning of ‘two-way identification based on the output embeddings of AlexNet’?
4. Although the article shows some details of the experiments, there are still some that have been ignored, such as the optimizer and the encoder structure when calculating the brain score.
5. It would be better to show on which datasets the pre-trained methods such as the diffusion model were pre-trained.
6. What is the meaning of '-' in Table 5? Can't it be calculated? Some words are written inconsistently, such as trained and pre-trained.

**Questions:**

1. Why choose ridge regression and MLP to encode fMRI data instead of CNN? Convolution of images is a state-of-the-art method.
2. What is the low level in Table 3 and Table 4?
3. What is the meaning of ‘two-way identification based on the output embeddings of AlexNet’?
4. Although the article shows some details of the experiments, there are still some that have been ignored, such as the optimizer and the encoder structure when calculating the brain score.
5. It would be better to show on which datasets the pre-trained methods such as the diffusion model were pre-trained.
6. What is the meaning of '-' in Table 5? Can't it be calculated? Some words are written inconsistently, such as trained and pre-trained.

---

### Official Review · Reviewer_tdBX · 2024-11-11

**Soundness:** 2
**Presentation:** 3
**Contribution:** 2
**Rating:** 5
**Confidence:** 5

**Summary:**

This study introduces FitFovea, a brain-to-image reconstruction pipeline that aligns more closely with human visual processing by reconstructing foveated images from fMRI data instead of the original images. The approach has three main steps: generating pseudo-foveated images, decoding fMRI signals to foveated images, and then using these to create stimulus-related images. Rather than aiming for perfect pixel accuracy, FitFovea emphasizes capturing perceptually-aligned visual details. The study also introduces a brain score metric to validate its effectiveness, showing that these foveated images can be more accurately decoded then their original high-resolution counterparts

**Strengths:**

- The paper raises an important question of what can in principle be decodeable from brain signals given the limited capacity of perceptual experience; this question has received little attention in prior decoding studies. Grounding the approach in human vision’s focus on central details adds strong cognitive motivation.
- The authors perform extensive evaluation of the reconstructed images against existing models using multiple metrics
- They incorporate state-of-the-art techniques for both saliency-based foveated image synthesis and generative image reconstruction from fMRI (eg Stable Diffusion), pushing the technical boundaries in brain-to-image decoding.

**Weaknesses:**

- My major concern is about whether decoding success with foveated images could be attributed to their reduced information content, making it inherently easier to match patterns in brain signals. To address this, a useful control would be to compare foveated images with a set of matched, low-pass filtered versions of the natural images that have similar information content but lack the specific spatial emphasis of foveation.
- Further, the difference between foveated and normal images appears only marginal based on the proposed brain score metric. Error bars would be helpful here. Why not use standard encoders commonly used in brainscore style studies to compare natural vs foveated images?
- Also, the rationale behind the third step in the pipeline (the stimulus generation phase) is unclear - is it used only to compare against existing methods? The original idea in the paper that brain signals would not be able to reconstruct inputs at high fidelity due to the foveation conflicts with this phase

**Questions:**

My biggest question is regarding the first concern above -
I'd suggest trying a control experiment where you generate low-pass filtered images that reduce high-frequency details, similar to how foveation reduces peripheral details, but without concentrating detail in the foveal region. This way, you’d control for overall information content without introducing a foveated structure. Would it be easier to deocde these images, similar to how its easy to decode foveated images?
If decoding is better for foveated images specifically (and not for the low-pass filtered ones), this suggests that foveation itself, not simply lower information content, may align better with how visual information is processed or represented in the brain

---

### Note · Authors · 2024-11-14

I have read and agree with the venue's withdrawal policy on behalf of myself and my co-authors.